# Current Understanding of bHLH Transcription Factors in Plant Abiotic Stress Tolerance

**DOI:** 10.3390/ijms22094921

**Published:** 2021-05-06

**Authors:** Jianrong Guo, Baixue Sun, Huanrong He, Yifan Zhang, Huaying Tian, Baoshan Wang

**Affiliations:** 1Shandong Provincial Key Laboratory of Plant Stress, College of Life Science, Shandong Normal University, Ji’nan 250014, China; a19106443425@163.com (B.S.); a19862189686@163.com (H.H.); skyskyskz@163.com (Y.Z.); 2College of Forestry Engineering, Shandong Agriculture and Engineering University, Ji’nan 250100, China; tianhy0312@163.com

**Keywords:** abiotic stress, gene regulation, bHLH, transcription factor, tolerance

## Abstract

Named for the characteristic basic helix-loop-helix (bHLH) region in their protein structure, bHLH proteins are a widespread transcription factor class in eukaryotes. bHLHs transcriptionally regulate their target genes by binding to specific positions on their promoters and thereby direct a variety of plant developmental and metabolic processes, such as photomorphogenesis, flowering induction, shade avoidance, and secondary metabolite biosynthesis, which are important for promoting plant tolerance or adaptation to adverse environments. In this review, we discuss the vital roles of bHLHs in plant responses to abiotic stresses, such as drought, salinity, cold, and iron deficiency. We suggest directions for future studies into the roles of *bHLH* genes in plant and discuss their potential applications in crop breeding.

## 1. Introduction

Abiotic stresses are major factors inhibiting crop production and reducing crop yields worldwide [1]. These stresses, including extreme temperatures (heat, chilling, and freezing), extreme water levels (flooding and drought), heavy metals, high pH, and salinity [2,3,4], influence vital biological factors in plants, such as nutrient supply, material biosynthesis, metabolism, and energy supply. If a plant cannot acclimate to abiotic factors that exceed a threshold level, it will experience stress; therefore, to ensure survival in the given conditions, plants employ many acclimatization mechanisms. Different signaling pathways can make plants more tolerant of stresses and allow them to continue to grow under stress conditions; however, the challenge facing researchers is to decipher how the signaling pathways in a specific plant species will acclimate to a given environmental stress.

The stress responses and tolerance levels of plants are inseparable from their expression of specific sets of genes. The basic helix-loop-helix (bHLH) transcription factors (TFs) play important roles in the stress tolerance of plants, in addition to their important roles in reproduction, such as in flower and fruit development [5], and in the biosynthesis of secondary metabolites, such as anthocyanin [6]. However, most reports on bHLH TFs focus on a small subset of model and crop plants. Although a variety of molecular biological methods have been used to explore the roles of bHLH TFs at various stages of plant growth and development and in metabolic pathways and stress responses, much remains to be learned about the functions and regulatory roles of these intriguing proteins. Here, we focus on recent studies of the roles of bHLHs in the molecular mechanisms that regulate abiotic stress tolerance in plants. These studies lay the foundation for improving the stress tolerance of crops.

## 2. Plant *bHLH* Genes

The bHLH family is the second largest family of eukaryotic TFs after the MYBs and is widely found in plants. The first plant *bHLH* gene to be reported was identified in maize (*Zea mays* L.), and 162 *bHLH* genes have been identified in the model plant *Arabidopsis thaliana* ((L.) Heynh. [7]. Various *bHLH* gene numbers have been reported in different plant species; for example, 122 *bHLH* genes have been identified in pepper (*Capsicum annuum* L.), which were categorized into 21 subfamilies based on their position-specific conserved amino acids and the presence of other conserved structural domains [8]. In rice (*Oryza sativa* L.), 167 *bHLH* family members were identified and classified into 22 subfamilies [9]. In potato (*Solanum tuberosum* L.), 124 *bHLH* genes were identified during a whole-genome analysis [10], while in cucumber (*Cucumis sativus* L.), 142 *bHLH* genes were identified and classified into 32 subfamilies [11]. In the common bean (*Phaseolus vulgaris* L.), 155 *bHLH* genes were identified and classified into 21 subfamilies [12].

The functions of *bHLH* genes have been analyzed in various biological processes in a variety of flowering plants; for example, 261 *bHLH* genes were identified in the peanut (*Arachis hypogaea* L.) genome, and they were assessed for their involvement in pod development [13]. A total of 137 *bHLH* genes in 26 subfamilies were identified in Jilin ginseng (*Panax ginseng* C. A. Mey.), some of which were found to function in the saline stress response [14]. In pear (*Pyrus bretschneideri* Rehder), 197 *bHLH* genes were identified (classified into 21 groups), with most playing essential roles in stress tolerance [15]. In addition, several other Rosaceae species were analyzed, including peach (*Prunus persica* (L.) Batsch), apple (*Malus×domestica* Borkh.), Chinese plum (*Prunus mume* Siebold & Zucc.), and strawberry (*Fragaria vesca* L.), for which 129, 188, 122, and 112 *bHLH* genes were identified, respectively, with many found to function in the stress responses of these plants [16]. Many of the *bHLH* genes detected in major crops such as rice, wheat (*Triticum aestivum* L.), and maize were shown to be involved in the responses to abiotic stresses [17]. In *Brassica napus* L., 460 *bHLH* genes were identified, which display different expression patterns between different organs, such as the roots and leaves, which may be related to the subfunctionalization of genes [18]. Although these studies represent a good foundation for understanding the roles of *bHLHs* in plants, the biological and regulatory functions of most of these TFs require further detailed study.

## 3. Roles of bHLHs in Plant Growth and Development

bHLH TFs participate in many growth and development processes, including seed germination and the development of carpels, anthers, epidermal cells, stomata, and fruit [19]. The Arabidopsis bHLH PHYTOCHROME-INTERACTING FACTOR 1 (PIF1) is a regulator of chlorophyll biosynthesis, which inhibits seed germination in the dark. Light induces the degradation of PIF1 and thus promotes photomorphogenesis [20]. PIF1 degradation is related to phyB activity [21]. In wounded poplar (*Populus deltoids*) plants, the biosynthesis of anthocyanin, a pigment important in plant defense, is induced via a pathway regulated by PdTT8 (a bHLH transcription factor) together with PdMYB118 (a MYB transcription factor) [22]. Additionally, bHLH TFs play important roles in anthocyanin accumulation in Arabidopsis, which is associated with the jasmonate (JA)-regulated plant defense pathway [23]. CRYPTOCHROME-INTERACTING BASIC HELIX-LOOP-HELIX 1 (CIB1) functions, together with the cryptochromes (CRYs), to promote flower initiation and development by stimulating the expression of flowering locus genes [24,25]. This is the first example of a heterodimeric action by the plant bHLH TFs that alters their DNA-binding affinity or specificity. Subsequently, other types of CIBs were found to jointly regulate the flowering time of Arabidopsis by activating the FLOWERING LOCUS T (*FT*) promoter and positively regulating the CRY2-mediated flowering pathway [26]. Additionally, bHLH TFs function in the regulation of plant cell elongation during the process of shade avoidance [27].

## 4. Roles of Plant bHLHs in Biosynthetic Processes

In addition to their roles in growth and development, bHLH TFs play vital roles in plant biosynthetic processes, such as anthocyanin biosynthesis. In peony (*Paeonia suffruticosa* Andrews) for example, anthocyanin biosynthesis in the flowers is positively regulated by PsbHLH1, which can directly bind to the promoters of dihydroflavonol 4-reductase (*PsDFR)* and anthocyanidin synthase (*PsANS)* genes to transcriptionally activate their expression, and could therefore be expected to be used to breed novel color cultivars [28]. In mulberry (*Morus alba* L.), *bHLH3* is a key gene in the regulation of fruit color formation; the pigment composition of mulberry fruits is disrupted by the altered expression of *bHLH3* [29]. By contrast, the transgenic expression of the wintersweet (*Chimonanthus praecox* L.) gene *CpbHLH1* inhibits the accumulation of anthocyanin in Arabidopsis [6]. Additionally, members of the *bHLH* family participate in terpenoid biosynthesis in *Medicago truncatula* (Gaertn.); the enhanced expression of TRITERPENE SAPONIN BIOSYNTHESIS-ACTIVATING REGULATOR 1 (*TSAR1*) and *TSAR2* (two JA-induced *bHLH* genes), together with the upregulation of downstream biosynthetic genes, results in an elevated triterpene saponin content [30]. The carbohydrate and malate accumulation in apples is regulated by the bHLH MdbHLH3 [31]. In *Artemisia annua* (L.), artemisinin biosynthesis is positively regulated by *AabHLH1* [32]. Amygdalin biosynthesis is also regulated by bHLH TFs in almond (*Prunus dulcis* Miller (D. A. Webb), syn. *Prunus amygdalus* L.) [33]. Biosynthetic functions for more *bHLH* genes will likely be identified [34].

## 5. Roles of bHLH TFs in Plant Stress Tolerance

In addition to the roles of the bHLHs in normal plant growth, development [35], flowering [36], and metabolic biosynthesis [28], many bHLHs function in signal transduction and the response to biotic or abiotic stresses, such as salinity, drought, low temperature, and nutrient deficits [37].

### 5.1. Roles of bHLH in Drought Tolerance

Drought is a major abiotic stress that limits plant growth and survival, crop quality, and production. *bHLH* genes respond to drought stress and enhance plant tolerance to water limitation; for example, *MdbHLH130* in apple improves the water deficit stress tolerance of transgenic tobacco (*Nicotiana. tabacum* L. “NC89”) by maintaining reactive oxygen species (ROS) homeostasis and inducing stomatal closure [38]. The transgenic expression of *PebHLH35* from desert poplar (*Populus euphratica* Olivier) in Arabidopsis increases tolerance to water deficit stress by regulating stomatal development and photosynthesis in the resulting plants [39]. A genomic analysis in foxtail millet (*Setaria italica* (L.) P. Beauv.) indicates that many *bHLH* genes function in drought tolerance [40].

The bHLH-induced enhancement of plant drought tolerance is generally related to abscisic acid (ABA) signaling; for example, *bHLH122*, which is highly expressed in guard cells, could enhance drought stress tolerance in Arabidopsis by repressing the catabolism of ABA, thus increasing the ABA content [41]. In wheat, drought adaptability is improved by the regulation of the ABA pathway by the *TabHLH1* gene [42]. Similarly, drought and ABA treatments increase the expression of *PebHLH35* in desert poplar, indicating that this gene is involved in the ABA pathway [39]. In rice, the over-expression of *OsbHLH148*, which regulates the JA pathway and the function of the OsJAZ (jasmonate ZIM domain) protein, increases the drought tolerance of the plants [43,44]. The ability of water retention and drought tolerance in transgenic Arabidopsis is increased with the expression of *MfbHLH38* from the resurrection plant (*Myrothamnus flabellifolius* Welw.), as well as the increase of their osmotic regulatory ability and oxidative stress tolerance, which is associated with the elevated ABA content and ABA response [45]. A similar result was obtained for the *CsbHLH041* gene from cucumber, which enhances the drought tolerance of Arabidopsis and cucumber seedlings [11]. These results indicate that *bHLH* genes play important roles in plant drought tolerance that associate with phytohormone ABA or JA and ROS scavenging, which highlight new areas of interest for research into crop drought tolerance. Current studies have focused on hormone-regulated stomatal movement or ROS scavenging, and perhaps, other functions of the bHLH TFs involved in drought stress need to be further explored. A summary model of the roles of bHLHs in plants is displayed in Figure 1.

### 5.2. Roles of bHLHs in Salt Tolerance

Salinity adversely affects plant growth, reducing germination rates, plant vigor, and crop yields [4,46,47,48]. In China alone, over one million acres of agricultural soils are contaminated with salt [46,49,50]. Many *bHLH* genes are involved in plant tolerance to salinity injury and play important roles in improving salt tolerance.

In plants, the first bHLH shown to be involved in salt tolerance, the calcium-binding NaCl-inducible gene 1 (*AtNIG1* from *Arabidopsis thaliana*), was identified in Arabidopsis, with *AtNIG1* overexpressors displaying a higher salt tolerance than the corresponding knockout mutants [51]. Similarly, the overexpression of *AtbHLH122* confers an increased salt tolerance, osmotic-regulating capacity, and proline concentration [41]. The expression of another *bHLH* gene, *AtbHLH92*, is upregulated by salinity and drought [52]. The salt tolerance of Arabidopsis plants was enhanced with the expression of *AtMYC2* gene, which is activated by a mitogen-activated protein kinase (MAPK), and is associated with increasing the levels of proline [53]. The functions of *bHLH* genes have also been explored in non-model plants; for example, *CsbHLH041* from cucumber confers ABA-induced salt tolerance in Arabidopsis and cucumber seedlings [11]. The transgenic expression of *VvbHLH1* from grape (*Vitis vinifera* L.) significantly increases flavonoid accumulation and enhances drought tolerance in Arabidopsis [54], while the overexpression of *SlbHLH22* in tomato (*Solanum lycopersicum* L.) seedlings improves salt tolerance, enhances ROS scavenging and increases the osmotic adjustment potential [55]. Enhanced salt tolerance was also observed in Arabidopsis heterologously expressing *MfbHLH38* from *Myrothamnus flabellifolius* [45]. Ion transport is an important process in salt tolerance, and some bHLH TFs regulate the Na^+^/H^+^ antiporter NHX. For instance, the bHLH TFs AtMYC2 and AtbHLH122 are the upstream regulators of *AtNHX1* and *AtNHX6*, respectively, which enhance salt tolerance in Arabidopsis [56]. Salt tolerance is also enhanced in transgenic Arabidopsis expressing *OrbHLH001,* a gene from Dongxiang wild rice (*Oryza rufipogon* Griff.) that is induced by salinity [57], while in cultivated rice, the expression of *OsAKT1* (inward-rectifying K^+^ channel) is induced and the ionic balance is maintained by the expression of *OrbHLH001* under salt stress [58]. Additionally, in maize, *ZmbHLH55* improves salt stress tolerance by increasing the accumulation of ascorbic acid through the direct modulation of the expression of genes involved in ascorbic acid biosynthesis [59]. These results provide insights into the mechanisms by which members of the *bHLH* gene family respond to salt stress in plants (Figure 2). In fact, certain similarities exist between salt and drought stress; for example, water deficiency stress will be caused, and high ABA levels will be synthesized in plants under either salt or drought stress. While for salt stress, a series of secondary stresses will be caused. Perhaps, the roles of bHLH TFs in maintaining ion homeostasis or metabolic balance in plants under salt stress may be further focused on.

### 5.3. Roles of bHLH in Cold Stress

Low temperature is a major environmental factor that adversely affects plant growth, development, yield, quality, and the geographical distribution of crops. To respond to cold stress, plants must regulate various physiological and biochemical processes, which often requires bHLH TF activity. The transgenic expression of *VabHLH1* from the wild amur grape (*Vitis amurensis* Rupr.) enhances the cold tolerance of transgenic Arabidopsis [60], while in rice, *OsbHLH1* expression is specifically induced by the cold, indicating that this gene might participate in the cold signaling pathway [61]. Cold tolerance is enhanced in transgenic Arabidopsis plants heterologously expressing two *bHLH* genes (*VaICE1* (inducer of CBF (C-repeat binding factor) expression and *VaICE2*) from wild *Vitis amurensis* [62]. The cold tolerance of transgenic pummelo (*Citrus grandis* (L.) Osbeck) is enhanced by the expression of the trifoliate orange (*Poncirus trifoliata* (L.) Raf.) *PtrbHLH,* gene, which modulates H_2_O_2_ levels in plant [63]. In transgenic tobacco, chilling tolerance is enhanced by the expression of *MdCIbHLH1* from apple [64]. Furthermore, the transgenic expression of *FtbHLH2* in tartary buckwheat (*Fagopyrum tataricum* (L.) Gaertn.) seedlings is significantly induced by cold stress, and the cold tolerance of transgenic Arabidopsis plants is enhanced by the expression of *FtbHLH2* [65].

As the cold tolerance of trees directly affects their distribution, studying cold tolerance mechanisms could lay the foundation for expanding the range of tree species into colder regions. Several *bHLH* genes in sweet cherry (*Prunus avium* L.), such as *PavbHLH1*, *PavbHLH18*, *PavbHLH28*, *PavbHLH60*, *PavbHLH61*, *PavbHLH65*, and *PavbHLH66*, are preliminarily shown to be involved in the response to cold stress based on systematic analyses performed on the genome [66]. The *bHLH* gene *MdbHLH3* enhances the cold tolerance of apples by regulating the increased accumulation of anthocyanin in low temperatures [67]. The transgenic expression of *DlICE1* (inducer of CBF expression 1, a bHLH gene) from longan (*Dimocarpus longan* Lour.) enhances the cold tolerance of Arabidopsis by increasing the proline content and reducing the malondialdehyde content in the seedlings [68], which is similar to the effect in banana [69]. Thus, *bHLH* genes play important roles in plant cold tolerance that is associated with enhanced proline accumulation, reduced malondialdehyde contents, and reduced electrolyte leakage [70]. Possible regulatory mechanisms involved in bHLH-mediated cold tolerance are shown in Figure 3.

### 5.4. Roles of bHLH in Iron Deficiency Stress

As a vital micronutrient involved in many metabolic processes, iron is indispensable for plant growth and development [71]. Some *bHLH* genes have been shown to function in the maintenance of iron homeostasis in plants [72,73]. In Arabidopsis seedlings, AtbHLH105/IAA-LEUCINE RESISTANT 3 (*ILR3*) is involved in the regulation of iron balance [74], and other *bHLH* genes, such as *AtbHLH38, AtbHLH39*, *AtbHLH100, AtbHLH101*, and *AtbHLH115*, have been confirmed to play important roles in iron uptake under iron-deficient conditions [75,76]. Other roles of *bHLH* genes (*AtbHLH38* and *AtbHLH39*) were also identified in Arabidopsis to respond to copper deficiency [77]. In rice, the iron-related transcription factor 3 (*OsIRO3*) that belongs to the *bHLH* gene family, plays a critical role in maintaining iron homeostasis in an iron-deficient environment [78,79]. In chrysanthemum (chrysanthemum cultivar “Jinba”), iron uptake and H^+^-ATPase levels are enhanced by the expression of *CmbHLH1* in a low-iron environment [80]. Loss-of-function mutant studies reveal that *bHLH34* and *bHLH104* maintain the iron balance in Arabidopsis, with the mutants displaying a disrupted response to iron deficiency [81,82]. In soybean (*Glycine max* (L.) Merr.), two *bHLH* genes, *GmbHLH57* and *GmbHLH300*, participate in iron homeostasis under iron-restricted conditions [83]. In tobacco, the expression of *NtbHLH1* is induced under iron deficiency to confer a stress response [84]. In iron-restricted environments, the expression of *bHLH* genes, such as *AtbHLH29*, is essential for plants to absorb more iron [85]. *bHLH* genes also play vital roles in responding to iron-deficient conditions and regulating iron uptake and balance in fruit trees, such as pummelo (*Citrus grandis* L.) [86]. These results provide insights that could enable the improvement of iron absorption and utilization in environments with low iron levels (Figure 4). Perhaps, this could provide us a basis for investigating the nutrient balance and elemental utilization, such as nitrogen and calcium, in plants under adversity stress.

Additionally, the functions of various investigated bHLH genes in plants related to abiotic stresses have been summarized and shown in Table 1.

## 6. Summary and Perspectives

The bHLH family of transcription factors has been extensively studied, with increasing attention being paid to their roles as key factors in the responses to environmental stressors, such as low temperatures, salinity, drought, iron or copper deficiency, and even low nitrogen. To date, however, most reports have mainly concentrated on a small subset of model and crop plants. Although some studies have explored the roles of *bHLH* genes in the growth, development, and stress responses of other plant species using a variety of molecular biological methods, further elucidation of their functions and regulatory roles is still required. Additionally, the roles of *bHLH* genes in other plants, such as halophytes, should be further explored. This review highlights several directions for future research that will reveal the roles of *bHLH* genes in plants and facilitate the usage of these genes in crops, particularly in efforts to breed crop varieties with improved stress tolerance, while the regulatory mechanisms of bHLH TFs still need to be investigated systematically.

## Figures and Tables

**Figure 1 ijms-22-04921-f001:**
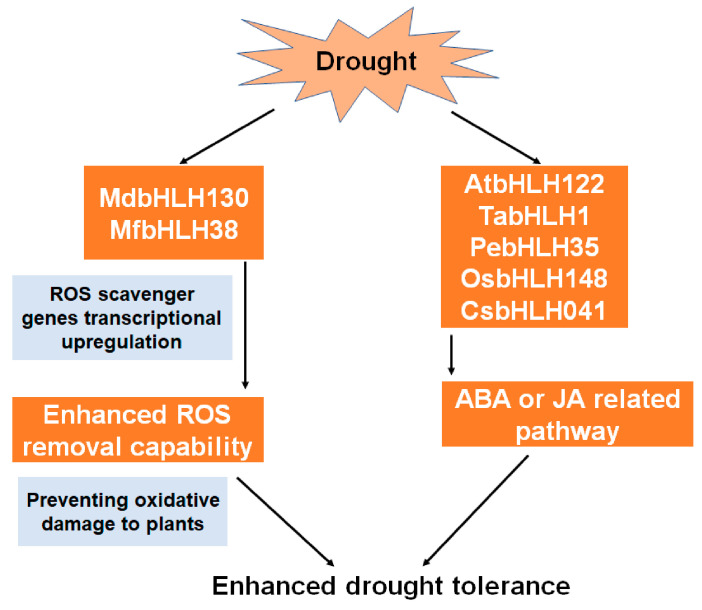
Model of the roles of bHLHs in drought tolerance.

**Figure 2 ijms-22-04921-f002:**
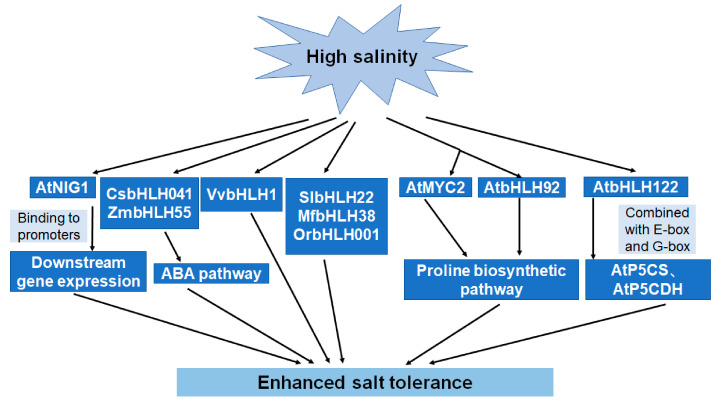
Model of the roles of bHLHs in salt tolerance.

**Figure 3 ijms-22-04921-f003:**
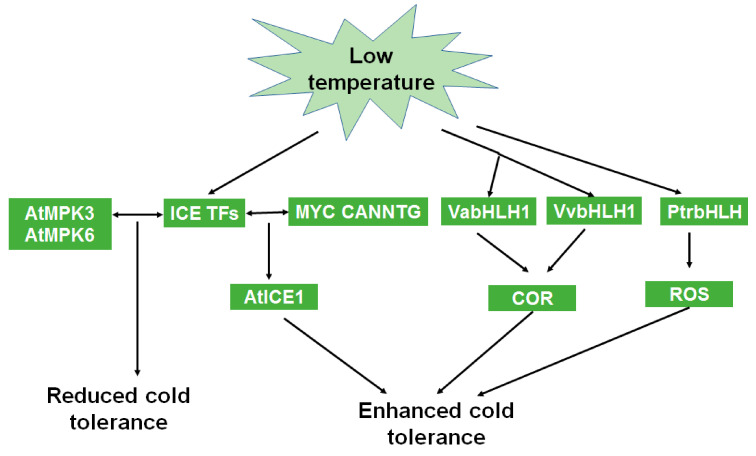
Model of the roles of bHLHs in cold stress tolerance.

**Figure 4 ijms-22-04921-f004:**
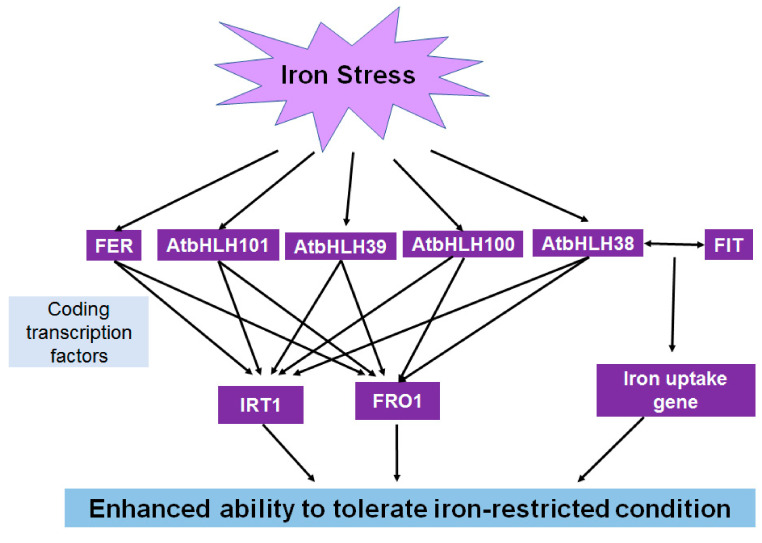
Model of the roles of bHLHs in iron stress tolerance.

**Table 1 ijms-22-04921-t001:** Studied basic helix-loop-helix (*bHLH*) genes that are involved in abiotic stress response in plants.

Gene	Function	Reference(s)
**Drought tolerance**		
*MdbHLH130*	Increase water deficit response and reactive oxygen species (ROS)-scavenging ability	[38]
*At* *bHLH122*	Express in guard cells and increase the abscisic acid (ABA) content	[41]
*TabHLH1*	Be involved in the ABA pathway	[42]
*PebHLH35*	Be involved in the ABA pathway	[39]
*OsbHLH148*	Be involved in the jasmonate (JA) pathway and increase drought tolerance	[43]
*MfbHLH38*	Increase osmotic regulation and oxidative stress tolerance ability	[45]
*CsbHLH041*	Enhance drought tolerance	[11]
**Salt tolerance**		
*AtNIG1*	Bind with calcium and enhance salt tolerance	[51]
*CsbHLH041*	Be associated with the ABA pathway and enhance salt tolerance	[11]
*AtbHLH122*	Be associated with proline accumulation and enhance salt tolerance	[41,56]
*AtbHLH92*	Response to osmotic stress and enhance salt and drought resistance	[52]
*AtMYC2*	Enhance the proline level and salt tolerance	[53]
*VvbHLH1*	Be involved in flavonoid accumulation and enhance salt tolerance	[54]
*MfbHLH38*	Be associated with the ABA pathway	[45]
*SlbHLH22*	Be associated with ROS scavenging	[55]
*OrbHLH001*	Be associated with ionic balance and enhance salt tolerance	[58]
*ZmbHLH55*	Be involved in ABA biosynthesis and improve salt tolerance	[59]
**Cold tolerance**		
*VabHLH1*	Be involved in C-repeat binding factor (CBF) cold signaling pathway and enhance cold tolerance	[60]
*OsbHLH1*	Participate in the cold signaling pathway	[61]
*VaICE1, 2*	Be involved in the CBF cold signaling pathway	[62]
*PtrbHLH*	Modulate H_2_O_2_ levels	[63]
*MdCIbHLH1*	Upregulate *MdCBF2* expression through the CBF pathway	[64]
*FtbHLH2*	Reduce ROS accumulation and enhance cold tolerance	[65]
*PavbHLH1, 18, 28, 60, 61, 65, and 66*	Enhance cold tolerance	[66]
*MdbHLH3*	Be involved in anthocyanin accumulation and enhance cold tolerance	[67]
*DlICE1, RmICE1*	Enhance proline accumulation and reduce malondialdehyde content	[68,70]
**Iron homeostasis**		
*AtILR3,* *AtbHLH38, 39, 100, 101, 115*	Regulate iron balance	[74]
*AtbHLH38, 39*	Respond to copper deficiency	[77]
*OsIRO3*	Maintain iron homeostasis	[79]
*CmbHLH1*	Regulate iron uptake and H^+^-ATPase	[80]
*AtbHLH34, 104*	Maintain iron balance	[81]
*GmbHLH57, 300*	Participate in iron homeostasis	[83]
*NtbHLH1*	Respond to iron deficiency	[84]

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
