# Peer review of "Current Understanding of bHLH Transcription Factors in Plant Abiotic Stress Tolerance"

_ijms, 2021, doi:10.3390/ijms22094921_

Round 1

Reviewer 1 Report

The manuscript provides a straight-forward review of studies on bHLH transcriptions factors in plants, particularly those involved in abiotic stress. A similar review was published in 2018 (Sun et al. 2018). The language is good, the reference list appears sufficient though is not exhaustive. The claim that the authors provide suggestions for bHLH TFs to plant breeding is somewhat weak, but could be strengthened. The organization within the sections is sometimes haphazard. The manuscript generally summarizes studies in a list style, and provides little synthesis of studies. The figures provide a little synthesis of the data. There is some value in list-style reviews such as this, but the review might be improved by organizing the paragraphs based on mechanisms. For example, in the subsection on drought tolerance, they roughly organize the paragraphs by bHLH TFs that affect stomata, and that bHLH TFs that act through ABA. But they divide their Figure 1 into two categories: ROS removal and hormone signaling (I’m going to ignore here the SiWLIM2b in Fig 1 – I’m not sure why it is discussed as it appears not to be a bHLH family TF). If the text were organized into these categories, ROS removal and hormone signaling, this kind of mechanism-based organization could be helpful and make it a better review.  

Line 27, 29, 32: Please confirm you intend “adapt” rather than “acclimate”

Line 65-70: There appear to be no references for this sentence or the numbers on line 68?

Line 98-116: This is a recent important paper that perhaps deserves to be discussed in the section on biosynthetic processes: (Sánchez-Pérez et al. 2019)

Line 104: “could therefore be used to breed novel cultivar colors” This suggests a very common misconception among plant physiologists and geneticists that breeders have direct control over a single gene. They might with the use of CRISPR technology, but my experience is this technology is being generally avoided by breeders because consumers view it as “genetic engineering.” This would be a stronger example if there were evidence that changes in this TF result in different flower colors.

Lines 122- 150 : I suggest referring to “low water stress” or “water deficit stress” rather than “water stress,” because you refer to flooding stress less directly in the introduction. “Water stress” could potentially be interpreted as high water stress or flooding.

Line 131: “Indeed, drought tolerance was enhanced in transgenic rice that heterologously expressed SiWLIM2b from foxtail millet” – but this gene is a LIM family gene, with MYB elements, how does that relate to this review on bHLH TFs?

Line 140: Do you mean: the over-expression of OsbHLH148 increased drought tolerance?

Line 140: This paragraph begins by discussing the role of bHLH TFs in ABA signaling, and at line 140 transitions into bHLH TFs in JA signaling. The final sentences list bHLH TFs and organisms, but don’t appear to tie into ABA or JA signaling. The organization could be improved for clarity.

Line 143-145: Sentence is awkward, please reword.

Fig 1: In the gray box, I believe you actually mean ROS scavenger gene transcriptional regulation, rather than ROS transcriptional regulation – or feel free to use another term than ROS scavengers, but it’s not the ROS themselves that are upregulated. Also, would it be worth having separate boxes for ABA and JA? The mechanisms within this figure are not always reflected in the text too – for example CsbHLH041 promotes ROS removal in the figure, but there is no mechanism discussed in the text.

Line 149 or lines 154-157: Might be worth pointing out that salt and drought stress create osmotic stress and thus can be similar

Line 161: Suggest “Similarly,…”

Line 163-165: Sentence is awkward, please reword.

Line 171: Suggest “enhanced” and “increased” rather than “enhancing” and “increasing”

Fig 2: It is not clear which arrow the gray box reading “negative regulation” is intended to modify. I also wonder if it would be worth writing out “proline” in full rather than “pro

Line 196: Please confirm you mean “freezing tolerance” and not “cold tolerance.”

Line 216-219: Again, this paragraph starts out discussing trees, and then ends discussing tobacco and multiflora rosa. Moving this sentence would improve the organization.

Line 231-232: Sentence is awkward, please reword.

Line 224-247: My understanding was that iron deficiency is not a common problem unless the soil is too basic or too waterlogged. Why did you choose this sub-topic?

Line 245: Typo, pummelo?

I thank the authors for their well-written work. It was a pleasure to read and they brought a number of interesting studies to my attention.

References cited above:

Sánchez-Pérez R, Pavan S, Mazzeo R, et al (2019) Mutation of a bHLH transcription factor allowed almond domestication. Science (80- ) 364:1095–1098. https://doi.org/10.1126/science.aav8197

Sun X, Wang Y, Sui N (2018) Transcriptional regulation of bHLH during plant response to stress. Biochem Biophys Res Commun 503:397–401. https://doi.org/10.1016/j.bbrc.2018.07.123

Author Response

Response to Reviewer 1 Comments

Dear Reviewer,

Thank you very much for your critical reading and professional comments of the manuscript. Those comments are very valuable and helpful for revising and improving our paper, as well as the important guiding significance to our future researches. We have studied comments carefully and have made correction point by point which we hope meets with approval. Revised parts are marked in blue in the revised paper. The following is the detailed explanation how we complied with the reviewers’ suggestions.

Point 1: The manuscript provides a straight-forward review of studies on bHLH transcriptions factors in plants, particularly those involved in abiotic stress. A similar review was published in 2018 (Sun et al. 2018). The language is good, the reference list appears sufficient though is not exhaustive. The claim that the authors provide suggestions for bHLH TFs to plant breeding is somewhat weak, but could be strengthened. The organization within the sections is sometimes haphazard. The manuscript generally summarizes studies in a list style, and provides little synthesis of studies. The figures provide a little synthesis of the data. There is some value in list-style reviews such as this, but the review might be improved by organizing the paragraphs based on mechanisms. For example, in the subsection on drought tolerance, they roughly organize the paragraphs by bHLH TFs that affect stomata, and that bHLH TFs that act through ABA. But they divide their Figure 1 into two categories: ROS removal and hormone signaling (I’m going to ignore here the SiWLIM2b in Fig 1 – I’m not sure why it is discussed as it appears not to be a bHLH family TF). If the text were organized into these categories, ROS removal and hormone signaling, this kind of mechanism-based organization could be helpful and make it a better review.  

Response 1: Thank you very much for your suggestion. Yes, you are right. In the drought tolerance of plants, bHLH TFs that will regulate stomata, and this pathway is correlated to ABA. Results also showed that the ROS homeostasis in plant during drought tolerance was regulated by bHLH TFs, such as MdbHLH130 from apple, MfbHLH38 from the resurrection plant (Myrothamnus flabellifolius Welw.). And We have deleted the SiWLIM2b related content and the reference in the figure and in the manuscript.

Point 2: Line 27, 29, 32: Please confirm you intend “adapt” rather than “acclimate”

Response 2: Thank you very much. We have changed “adapt” into “acclimate” in the manuscript according to your suggestion.

Point 3: Line 65-70: There appear to be no references for this sentence or the numbers on line 68?

Response 3: So sorry for that, there should be a reference here, which is missing, we have already added it to the manuscript and in the reference list.

Point 4: Line 98-116: This is a recent important paper that perhaps deserves to be discussed in the section on biosynthetic processes: (Sánchez-Pérez et al. 2019)

Response 4: Thank you for your suggestion. We have added the corresponding content of the reference Sánchez-Pérez et al. 2019 in the manuscript and in the reference list.

Point 5: Line 104: “could therefore be used to breed novel cultivar colors” This suggests a very common misconception among plant physiologists and geneticists that breeders have direct control over a single gene. They might with the use of CRISPR technology, but my experience is this technology is being generally avoided by breeders because consumers view it as “genetic engineering.” This would be a stronger example if there were evidence that changes in this TF result in different flower colors.

Response 5: Thank you very much. Yes, you are right, phenotype in plant, such as color, will not be controlled by a single gene. In this process, there may be many other genes were involved in. We have modified the sentence to “…and could therefore be expected to be used to breed novel color cultivars.” in the manuscript.

Point 6: Lines 122- 150 : I suggest referring to “low water stress” or “water deficit stress” rather than “water stress,” because you refer to flooding stress less directly in the introduction. “Water stress” could potentially be interpreted as high water stress or flooding.

Response 6: Thank you for your suggestion. We have changed “water stress” into “water deficit stress” in the manuscript.

Point 7: Line 131: “Indeed, drought tolerance was enhanced in transgenic rice that heterologously expressed SiWLIM2b from foxtail millet” – but this gene is a LIM family gene, with MYB elements, how does that relate to this review on bHLH TFs?

Response 7: Thank you for your suggestion. Yes, you are right. We have deleted the SiWLIM2b related content and the reference in the figure and in the manuscript.

Point 8: Line 140: Do you mean: the over-expression of OsbHLH148 increased drought tolerance?

Response 8: Thank you for your suggestion. Yes, you are right. We have changed it to “over-expression” in the manuscript.

Point 9: Line 140: This paragraph begins by discussing the role of bHLH TFs in ABA signaling, and at line 140 transitions into bHLH TFs in JA signaling. The final sentences list bHLH TFs and organisms, but don’t appear to tie into ABA or JA signaling. The organization could be improved for clarity.

Response 9: Thank you for your suggestion. We have changed the sentence to “These results indicate that bHLH genes play important roles in plant drought tolerance that associated with phytohormone ABA or JA, and ROS scavenging, which highlight new areas of interest for research into crop salt tolerance.” in the manuscript.

Point 10: Line 143-145: Sentence is awkward, please reword.

Response 10: Thank you for your suggestion. We have changed the sentence to “The ability of water retention and drought tolerance in transgenic Arabidopsis was increased with expression of MfbHLH38 from the resurrection plant (Myrothamnus flabellifolius Welw.), as well as the increased of their osmotic regulatory ability and oxidative stress tolerance, which is associated with the elevated ABA content and ABA response.” in the manuscript.

Point 11: Fig 1: In the gray box, I believe you actually mean ROS scavenger gene transcriptional regulation, rather than ROS transcriptional regulation – or feel free to use another term than ROS scavengers, but it’s not the ROS themselves that are upregulated. Also, would it be worth having separate boxes for ABA and JA? The mechanisms within this figure are not always reflected in the text too – for example CsbHLH041 promotes ROS removal in the figure, but there is no mechanism discussed in the text.

Response 11: Thank you for your suggestion. We have modified the figure, and Figure 1 has been replaced in the manuscript.

Point 12: Line 149 or lines 154-157: Might be worth pointing out that salt and drought stress create osmotic stress and thus can be similar

Response 12: Thank you. Yes, osmotic stress could be created under both salt and drought stress and thus could be similar. We have it edited to “crop drought tolerance” in the manuscript.

Point 13: Line 161: Suggest “Similarly,…”

Response 13: Thank you. Done.

Point 14: Line 163-165: Sentence is awkward, please reword.

Response 14: Thank you for your suggestion. We have changed the sentence to “The salt tolerance of Arabidopsis plants was enhanced with the expression of AtMYC2 gene, which is activated by a mitogen-activated protein kinase (MAPK), and is associated with increasing the levels of proline [54].” in the revised manuscript.

Point 15: Line 171: Suggest “enhanced” and “increased” rather than “enhancing” and “increasing”

Response 15: Thank you for your suggestion. Done.

Point 16: Fig 2: It is not clear which arrow the gray box reading “negative regulation” is intended to modify. I also wonder if it would be worth writing out “proline” in full rather than “pro

Response 16: Thank you for your suggestion. We have modified the Figure 2 in the revised manuscript.

Point 17: Line 196: Please confirm you mean “freezing tolerance” and not “cold tolerance.”

Response 17: Thank you for your suggestion. We have changed “freezing tolerance” to “cold tolerance.” in the revised manuscript.

Point 18: Line 216-219: Again, this paragraph starts out discussing trees, and then ends discussing tobacco and multiflora rosa. Moving this sentence would improve the organization.

Response 18: Thank you for your suggestion. We have moved and changed the sentence to “Thus, the bHLH genes play important roles in plant cold tolerance that was associated with enhanced proline accumulation, reduced malondialdehyde contents, and reduced electrolyte leakage [71]. Possible regulatory mechanisms involved in bHLH-mediated cold tolerance are shown in Figure 3.” in the revised manuscript.

Point 19: Line 231-232: Sentence is awkward, please reword.

Response 19: Thank you for your suggestion. We have changed the sentence to “Other roles of bHLH genes (AtbHLH38 and AtbHLH39) were also identified in Arabidopsis that in responding to copper deficiency [78]. In rice, the iron-related transcription factor 3 (OsIRO3) that belonged to bHLH gene family, plays a critical role in maintaining iron homeostasis in an iron-deficient environment [79, 80].” in the revised manuscript.

Point 20: Line 224-247: My understanding was that iron deficiency is not a common problem unless the soil is too basic or too waterlogged. Why did you choose this sub-topic?

Response 20: Thank you very much. Under saline-alkali conditions, the accumulation of a large amount of salt ions affects the absorption of other nutrients by plants, which will cause element deficiency. Iron is an element that plants will lack in saline-alkali conditions, so it is listed for discussion.

Point 21: Line 245: Typo, pummelo?

Response 21: Thank you very much. So sorry for that, we have changed it in the revised manuscript.

I thank the authors for their well-written work. It was a pleasure to read and they brought a number of interesting studies to my attention.

References cited above:

Sánchez-Pérez R, Pavan S, Mazzeo R, et al (2019) Mutation of a bHLH transcription factor allowed almond domestication. Science (80- ) 364:1095–1098. https://doi.org/10.1126/science.aav8197

Sun X, Wang Y, Sui N (2018) Transcriptional regulation of bHLH during plant response to stress. Biochem Biophys Res Commun 503:397–401. https://doi.org/10.1016/j.bbrc.2018.07.123

Reviewer 2 Report

The review manuscript entitled "Current understanding of bHLH transcription factors in plant abiotic stress tolerance" by Guo and collaborators appeared well conceived and one of the first review comprehensive of many, rather all, reports on the role of the important TF family bHLH in plant responses to abiotic stress. 

The manuscript is well organized and well written except some sentences in which the English style could be significantly improved by an English mother tongue; for this I strongly recommended the Authors to revise their manuscript.

I encourage the Authors to modify some aspects of each important sections of the review:

  1. I request to more focus and detail the report of such more important researches in each section to avoid the feeling that it is a mere list of references more or less;
  2. The figures comprehension could be improved by a detailed legend accompanied by some more information included in some sections of the figure;
  3. I warmly request one table (included in the manuscript) for each figure that reported the most important components included in each complex trait discussed with another column reported the name of the bHLH member involved and another ones reporting the reference already included in the manuscript.   

Once the authors have made the required changes the manuscript can be accepted for publication on Int. J. Mol. Science.                                   

Author Response

Response to Reviewer 2 Comments

Dear Reviewer,

Thank you very much for your critical reading and professional comments of the manuscript. Those comments are very valuable and helpful for revising and improving our paper, as well as the important guiding significance to our future researches. We have studied comments carefully and have made correction point by point which we hope meets with approval. Revised parts are marked in blue in the revised paper. The following is the detailed explanation how we complied with the reviewers’ suggestions.

The review manuscript entitled "Current understanding of bHLH transcription factors in plant abiotic stress tolerance" by Guo and collaborators appeared well conceived and one of the first review comprehensive of many, rather all, reports on the role of the important TF family bHLH in plant responses to abiotic stress. 

The manuscript is well organized and well written except some sentences in which the English style could be significantly improved by an English mother tongue; for this I strongly recommended the Authors to revise their manuscript.

I encourage the Authors to modify some aspects of each important sections of the review:

Point 1:I request to more focus and detail the report of such more important researches in each section to avoid the feeling that it is a mere list of references more or less;

Response 1: Thank you very much. We have modified the corresponding content in the revised manuscript according to your suggestion.

Point 2:The figures comprehension could be improved by a detailed legend accompanied by some more information included in some sections of the figure;

Response 2: Thank you very much. We have modified the figures in the sections of drought and salt stress in the revised manuscript according to your suggestion.

Point 3:I warmly request one table (included in the manuscript) for each figure that reported the most important components included in each complex trait discussed with another column reported the name of the bHLH member involved and another ones reporting the reference already included in the manuscript.   

Response 3: Thank you very much. We have added an integrated table for the figures in the revised manuscript according to your suggestion.